# A Preliminary Study of the Potential Molecular Mechanisms of Individual Growth and Rumen Development in Calves with Different Feeding Patterns

**DOI:** 10.3390/microorganisms11102423

**Published:** 2023-09-28

**Authors:** Jie Wang, Kaisen Zhao, Mianying Li, Huimei Fan, Meigui Wang, Siqi Xia, Yang Chen, Xue Bai, Zheliang Liu, Jiale Ni, Wenqiang Sun, Xianbo Jia, Songjia Lai

**Affiliations:** 1Farm Animal Genetic Resources Exploration and Innovation Key Laboratory of Sichuan Province, Sichuan Agricultural University, Chengdu 611130, China; wjie68@163.com (J.W.); wqsun2021@163.com (W.S.); jaxb369@sicau.edu.cn (X.J.); 2College of Animal Science and Technology, Sichuan Agricultural University, Chengdu 611130, China; zhaofl0303@163.com (K.Z.); lmyazure@163.com (M.L.); fanhuimei1998@163.com (H.F.); xiasiqi2020@163.com (S.X.);

**Keywords:** calves, feeding pattern, rumen, metagenomics, untargeted metabolomics

## Abstract

At present, it is common to feed calves with “Concentrate”, “Concentrate + hay” and TMR “Total Mixed Rations” feeding patterns in China, which achieved well feeding efficiency, but the three feeding patterns molecular regulation mechanism in actual production is still unclear. The study aimed to explore the most suitable feeding pattern for Chinese Holstein calves to improve the rumen fermentation function and growth performance of calves. In this regard, the interactions between rumen microorganisms and host metabolism were investigated. The rumen volume and weight of calves in the GF group were significantly higher than those in the GFF and TMR groups (*p* < 0.05), and the rumen pH of calves in the GF group was 6.47~6.79. Metagenomics analysis revealed that the rumen microbiome of GF and GFF calves had higher relative abundances of *Methanobrevibacter*, *Methanosphaera*, and *Methanolacinia* (*p* < 0.05). *Prevotella multisaccharivorax* was significantly more abundant in the rumen of GF calves (*p* < 0.05), indicating that GF group calves had a stronger ability to ferment sugars. Notably, in the pyruvate metabolic pathway, phosphoenolpyruvate carboxylase was significantly up-regulated in GF calves compared with the TMR group, and pyruvate-phosphate dikinase was significantly down-regulated. Metabolomic results showed that Ursodeoxycholic acid was significantly up-regulated in GF calves, and most of the differential metabolites were enriched in Bile secretion pathways. The association analysis study found that the microorganisms of *Prevotella* and *Ruminococcaceae* might cooperate with the host, which was helpful for the digestion and absorption of lipids and made the calves have better growth. The three feeding modes had similar effects, but the ‘GF’ feeding pattern was more beneficial to the individual growth and ruminal development regarding ruminal morphology, contents physiology and microorganisms. Furthermore, the synergistic effect of rumen microorganisms and the host could more effectively hydrolyze lipid substances and promote the absorption of lipids, which was of great significance to the growth of calves.

## 1. Introduction

The health and biological functioning of livestock are often prioritized. In recent years, nutritional strategies have emerged. It has been proposed as a critical factor in improving the health status and welfare of animals as well as enhancing productivity and performance in livestock [1,2], and it has been well demonstrated that nutritional strategies could increase the level of production and improve the health of animals and products obtained from them. Feed type, feeding method and nutrition level are related to the rumen development of calves [3], and the growth and development of calves during lactation determine the performance of final milk production to a certain extent [4]. At present, the feeding patterns of calves during lactation are different in China, mainly including “concentrate”, “TMR” and “concentrate plus hay” feeding patterns. Feeding only concentrate might occasionally lead to a lower pH value of ruminal liquid, aggregation of rumen papilla, incomplete keratinization, and other symptoms in calves [5]. A few papers reported that due to the high moisture content of TMR, it is not easy to achieve the ideal growth effect, and it is easy to cause calves diarrhea [6]. Premature hay intake by calves might lead to insufficient energy intake of animals and diseases (such as hay belly) due to their rumen not fully developed and their ability to digest roughage was limited. Too late intake of hay might affect the digestion ability of calves to roughage in adulthood, which was not conducive to rumen development and optimal production performance [7,8,9]. However, the molecular mechanism of individual growth and rumen development of calves with three feeding patterns remains unclear.

Microorganisms in the rumen interact with the host, regulate the nutrient metabolism of the organism, participate in the material transport process, and play an important role in the organism’s immune media, which could feedback affect the feed utilization rate and the growth performance of the host [10,11]. The research showed that diet was used as an exogenous dietary signal to mediate the interaction between host and rumen microbiota [12]. It is believed that diet is the main factor affecting the composition of rumen microbial structure [13], and the change in feed nutrient level might cause an adaptive change in the rumen microbial community. Feeding beef cattle with a high concentrate-to-feed ratio significantly increases the number of *Streptococcus bovis*, *Megasphaera elsdenii* and *Prevotella bryantii* and reduces the relative abundance of cellulolytic bacteria [14]. Microbes in the rumen of ruminants mainly include Ruminococcus, Fibrobacteres, Prevotellariae, etc., and they are closely related to the host organism’s production performance and health status of the host organism [15].

Additionally, short-chain fatty acids (SCFAs), metabolites of rumen microorganisms, are the main energy source, have anti-inflammatory and immunomodulatory effects, and could strengthen the intestinal barrier [16]. In recent years, more and more studies have explored the metabolic differences of calves based on metabolomics [17,18,19]. A few papers identified serum metabolites associated with calf nutrition and healthy development [20].

The study aimed to explore the effects of three different feeding patterns of “GF”, “GFF”, and “TMR” on the growth, rumen development, rumen fluid short-chain fatty acids, rumen microbes and organism metabolism of calves, and finally, select the most suitable feeding pattern for calves, reveal their mechanisms at the molecular level, which to lay a theoretical foundation for further explaining the structure and function of rumen microorganism and its degradation mechanism to feed, and provide a theoretical basis for calf feeding.

## 2. Materials and Methods

### 2.1. Ethics Statement

The study was approved and conducted in accordance with the ethical standards of the Institutional Animal Care and Use Committee of the College of Animal Science and Technology, Sichuan Agricultural University, Sichuan, China.

### 2.2. Animals

The study selected nine 7-day-old Chinese Holstein bull calves with healthy, and the average body weight was 41.58 ± 0.79 kg. The nine calves were randomly divided into three groups of GFF, GF and TMR, which, following the completely random grouping principles, feed and water were provided *ad libitum*, and each individual was raised in a single pen and fed milk. Based on feeding milk, the GFF group was fed with a “ concentrate” pattern. In the GF group, 10% hay was added based on fine feed, in which the mixture ratio of hay was alfalfa grass: oat grass = 3:2, cut into 1–2 cm long and mixed evenly. The TMR group was fed with a “TMR” feed pattern. Other feeding and management methods are carried out according to the existing methods of Sichuan Xuebao Dairy Group Co., Ltd., and the body weight of the calves at 80 days old and slaughter them. The formula and nutritional composition of the diet are shown in Appendix A.

### 2.3. Sample Collection

The sampling time for calves was from 10:00 to 11:00 a.m. Calves fasted for 24 h, jugular vein blood collection and timely centrifugation (4 °C, 3000 r/min, 5 min) to obtain serum. After slaughter, the rumen contents of calves were quickly collected and transported in a 2 mL cryopreservation tube and placed in liquid nitrogen. Finally, they were stored in a refrigerator at −80 °C for subsequent experiments, and all preliminary sample processing was completed within seven days after sample collection. At the same time, the pH value of rumen contents was measured with a PHS-25C PH tester (Ostell Industrial Co., Ltd., Shanghai, China), and rumen weight and volume were determined.

### 2.4. DNA Extraction, Metagenomic Sequencing

According to the kit instructions, microbial DNA was extracted from rumen content samples (the samples were liquid) using the Tiangen Magnetic Bead Kit (Tiangen Biotech, Beijing, China). The purity and integrity of DNA were detected by 1% agarose gel electrophoresis. DNA was quantified using Qubit^®^ dsDNA Assay Kit in Qubit^®^ 2.0 Flurometer (Life Technologies, Foster City, CA, USA), and appropriate samples were taken into centrifuge tubes, dilute the sample with sterile water to an OD value between 1.8~2.0. Then, taking 1 μg of genomic DNA from the sample, a metagenomic library was constructed using NEBNext^®^ Ultra DNA Library Prep Kit for Illumina (NEB, Illumina Inc., San Diego, CA, USA) and sequenced using the Illumina HiSeq Xten platform. The library construction and sequencing work were performed by Beijing Novogene Technology Co., Ltd. Company (Beijing, China). Preprocessing the Raw Data obtained from the Illumina HiSeq sequencing platform using Readfq (V8, https://github.com/cjfields/readfq, accessed on March 2022) to remove reads with low-quality bases (the default quality threshold < 38) exceeding 40 bp, N bases reaching 10 bp, and the overlap between adapters exceeding 15 bp. Then, using bowtie2 software (version 2.2.4, http://bowtie-bio.sourceforge.net/bowtie2/index.shtml, accessed on March 2022) to filter out DNA interference that may originate from calves [21,22]. Clean Data for each sample was assembled and analyzed using MEGAHIT software [23,24] (v1.0.4-beta, https://github.com/voutcn/megahit, accessed on March 2022). MetaGeneMark software [21] (V2.10, http://topaz.gatech.edu/GeneMark/, accessed on March 2022) was used to predict the open reading region (ORF) of scaftigs ≥ 500 bp in each sample, and the information with length less than 100 nt in the prediction results were filtered [25,26]. Finally, the CD-HIT software [27,28] (V4.5.8, http://www.bioinformatics.org/cd-hit, accessed on March 2022) was used to de-redundant the ORF prediction results to obtain a non-redundant initial gene catalogue, using Bowtie2 (Bowtie2.2.4) Compare the Clean Data of each sample to the initial gene catalogue, calculate the number of reads of the genes aligned in each sample [29,30], and obtain the Unigenes for subsequent analysis [31].

### 2.5. Classification and Functional Annotation of Rumen Metagenomes

According to NCBI’s NR database (Version 2018-01-02, https://www.ncbi.nlm.nih.gov/, accessed on April 2022), using DIAMOND software [32] (V0.9.9.110, https://github.com/bbuchfink/diamond/, accessed on April 2022)for taxonomic assessment of rumen microbes in calves. The LCA algorithm (Applied in Megan software for systematic classification, version 4.0, https://en.wikipedia.org/wiki/Lowest_common_ancestor, accessed on April 2022) was used to determine the species annotation information of the aligned sequences and to obtain the abundance information of each sample at the Kingdom, phylum, genus, and species. Starting from the abundances at each taxonomic level, PCA [33] (RADE4 package, version 2.15.3) dimension reduction analysis was performed, followed by lefse analysis using lefse software [34] (v1.0, LDA score default to 3) to look for differential species between groups.

Unigenes were compared with the KEGG [35] database (version 2018-01-01, http://www.kegg.jp/kegg/, accessed on April 2022) and CAZy [36] databases (version 201801, http://www.cazy.org/, accessed on April 2022) by using DIAMOND software (v0.9.9.110, https://github.com/bbuchfink/diamond/, accessed on April 2022), and alignments were performed with parameters set to blastp, -e 1e-5 [29]. Then, the abundances at different functional levels were counted, starting from the abundances at each taxonomic level, annotated gene number statistics, relative abundance profile display, metabolic pathway comparison analysis, and component difference analysis using lefse.

### 2.6. Rumen Fluid SCFA and Serum Metabolomic Analysis

The extraction of short-chain fatty acids (SCFA) in rumen fluid and serum metabolites, data preprocessing, and quantitative and qualitative analysis of metabolites were completed by Nuohe Zhiyuan Technology Co., Ltd. (Beijing, China). The concentration of SCFA in the rumen was detected and calculated by the standard curve of acetic acid, propionic acid, butyric acid, isobutyric acid, valeric acid, isovaleric acid and caproic acid. Then, metaX [37] was used to convert the data, principal component analysis (PCA) and partial least squares discriminant analysis (PLS-DA) were performed, and the fold change (FC value) of metabolites between the two groups was calculated. The default criteria for screening differential metabolites were VIP > 1, *p* < 0.05 and FC ≥ 2 or FC ≤ 0.5. Next, the functions of different metabolites and metabolic pathway analysis by the KEGG database (https://www.kegg.jp/kegg/pathway accessed on May 2022).

### 2.7. Statistical Analysis

Statistical analysis was performed using SPSS 27.0 software (IBM, Chicago, IL, USA), and the significant difference analysis among the three groups were performed by one-way ANOVA. It was statistically significant when *p* < 0.05.

To determine the relationship between rumen microorganisms and serum metabolites, spearman correlation analysis was used to analyze the correlation between 30 microbial species with the highest abundance and serum metabolites by R software(v4.1.3, and R package “pheatmap” package in R was used to draw a clustering heat map.

## 3. Results

### 3.1. Growth Traits Characteristics of Calves

The results of growth performance and rumen development indexes among the three patterns of calves were showed in Table 1. The slaughter weight of calves in the GF and GFF groups was significantly greater than that in the TMR group (*p* < 0.05), and the weight gain of calves in the GF and GFF groups also increased but not significantly. The rumen weight and volume of calves in the GF group were significantly higher than those in the other two groups (*p* < 0.05), while the rumen pH of calves in the GFF group decreased significantly compared to GF and TMR groups (*p* < 0.05). It was worth noting that t the ratio of rumen to live weight and the ratio of rumen to total stomach weight of calves in the GFF group were higher than those in the other two groups, indicated that the rumen development status of this group of calves was better.

The dominant short-chain fatty acids in rumen fluid were acetic, propionic and butyric acids in the three groups, and the contents of propionic acid and valeric acid in the rumen of calves in the GF group were significantly higher than those in GFF and TMR groups (*p* < 0.05, Figure 1A). Therefore, the three different feeding patterns achieved different growth promotion effects, but the ‘GF’ feeding pattern was more helpful to the rumen development of calves. Calves in the TMR group were accompanied by slight diarrhea in the late test experiment (Figure 1B), and ‘hairy bulbs’ were found in the rumen of the GFF group (Figure 1C).

### 3.2. Analysis of the Rumen Metagenome

A total of 247,127,428.4 Mbp of raw data and the clean data of 110,839.92 Mbp were obtained. The GF, GFF and TMR groups had 12,215.79 Mbp, 12,345.66 Mbp and 12,385.19 Mbp average of clean data, respectively. Q20 and Q30 of each sample were above 97.50% and 93.50%. A total of 1,536,918 contiguous clonal lines (N50 length of 30,779 ± 139 bp) were generated, yielding 170,769 ± 18,589 per sample (mean ± standard error) after Metagename assembly (Appendix A). The rumen microbes consisted of 84.374% bacteria (320,295,748 sequences), 0.898% archaea (3,408,383 sequences), 0.014% eukaryotes (51,112 sequences), and 0.146% viruses (552,601 sequences) (Table 2).

The results of the domains of calf rumen microbiota under three different feeding patterns are shown in Appendix A. There was a significant difference in Archaea between GF and TMR (*p* < 0.05). Bacteria and Eukaryota were significantly different between GFF and TMR groups, while Eukaryota was not different (*p* > 0.05) between GF and TMR groups. Furthermore, there was no significant difference in microbial composition between GF and GFF at the domain level.

### 3.3. Compositional Characteristics and Taxonomic Differences of the Microbiome

The study found that the dominant bacterial phylum was independent of the three feeding patterns. *Bacteroidetes* (40.46 ± 3.02%) was dominant, followed by *Firmicutes* (32.99 ± 3.23%), *Proteobacteria* (2.20 ± 0.47%) and *Actinobacteria* (1.91 ± 0.53%) (Table 3). However, its relative abundance was different. Compared with the TMR group, the relative abundance of *Firmicutes* of GF calves was significantly increased, but *Bacteroidetes* was significantly decreased (*p* < 0.05). Similar changes were also observed in the GFF group, but not significantly (Figure 2A). The dominant genera were *Prevotella* (27.76 ± 2.04%), *Clostridium* (7.31 ± 2.94%), *Olsenella* (1.20 ± 0.36%) and *Butyrivibrio* (1.09 ± 0.17%) (Appendix A). At the species level, the analysis showed the relative abundances of *Prevotella multisaccharivorax* and *Prevotella* sp. *AGR2160* were significantly increased in the GF and GFF groups compared with the TMR group (*p* < 0.05). The relative abundance of *Ruminococcaceae flavefaciens*, *Prevotella* sp. *Tc 2-28*, *Prevotella brevis* and *Alistipes* sp. *CAG: 435* was significantly decreased in the GF group (*p* < 0.05) but not significantly decreased in GFF, and *Prevotella ruminicola* was significantly decreased in the GF and GFF groups (*p* < 0.05, Figure 2B).

For the *Archaea*, compared with the TMR group, the abundance of *Euryarchaeota* (99.61 ± 0.11%) was extremely significantly higher in both the GF and GFF groups (*p* < 0.01, Appendix A). The abundance of *Candidatus Woesearchaeota* and *Candidatus Bathyarchaeota* decreased significantly in GF and GFF (*p* < 0.05). At the genus level, except for *Caldivirga*, the relative abundance of other genera decreased in GF and GFF groups (*p* < 0.05, Appendix A). At the species level, the relative abundances of *Methanobrevibacter wolinii*, *Methanobrevibacter boviskoreani* and *Methanobrevibacter* sp. *AbM4* were significantly increased in GF calves, and the relative abundances of *Methanobrevibacter ruminantium*, *Methanobrevibacter olleyae* and *Methanobrevibacter* sp. *YE315* were significantly decreased compared with the TMR group (*p* < 0.05). The calves in the GFF group also had similar but not significant changes (Appendix A).

### 3.4. Functional Characteristics and Functional Differences of Microbiome

The results of the KEGG function of the rumen microbial community showed a total of 266 pathways at the third level in GF, GFF and TMR groups (Appendix A). They were respectively for ‘Metabolism’ (52.75 ± 1.00%), ‘Genetic Information Processing’ (22.06 ± 1.12%), ‘Environmental Information Processing’ (9.14 ± 0.22%), ‘Cellular Processes’ (7.54 ± 0.13%), ‘Human Diseases’ (5.77 ± 0.07%) and ‘Organismal Systems’ (2.75 ± 0.04%). At the second level, a total of 42 categories were observed, including ‘Carbohydrate metabolism’ (3.31 ± 0.06%), ‘Energy metabolism’ (1.95 ± 0.04%), ‘neurodegenerative diseases’ (0.04 ± 0.01%), ‘Amino acid metabolism’ (2.8 ± 0.09%), ‘Translation’ (2.31 ± 0.15%) and ‘Membrane transport’ (1.31 ± 0.05%) which were the most abundant. At the third level, compared with the TMR group, there were 18 extremely significant enriched third-level pathways (including 5 “metabolic” pathways, 2 “environmental information processing” pathways, 4 “cellular process” pathways, 3 “human disease” pathways, 1 “genetic information processing” pathway and 3 “Organismal Systems” pathways) in the GF group, and only five third-level pathways were extremely significantly enriched in the GFF group (*p* < 0.01), including “Type II diabetes mellitus”, “Human papillomavirus infection”, “Mineral absorption”, “Nitrogen metabolism” and “Viral carcinogenesis”. Only one Caprolactam degradation was significantly enriched in the GF group compared with the GFF group (*p* < 0.01, Figure 3).

For the differential KEGG modules involved, 16 GF-enriched and 25 GFF-enriched modules were identified compared to the TMR group (*p* < 0.01, Figure 4A,B). Nine extremely significantly enriched modules were found between the GF and GFF groups, including 3 GF-enriched modules and 6 GFF-enriched modules (*p* < 0.01, Figure 4C). Regarding energy metabolism, only and M00172 (C4-dicarboxylic acid cycle) was enriched in the GF calves compared with GFF calves. However, about carbohydrate metabolism, only M00010 (oxaloacetate ≥ 2-oxoglutarate) among the above extremely significantly enriched in the GFF group, six modules enriched in the TMR group (LDA > 3 and *p* < 0.05, Figure 4D).

For the CAZymes map, a total of 278 genes encoding CAZymes were identified (Appendix A), including 124 glycoside hydrolases (GHs), 56 glycosyltransferases (GTs), 17 polysaccharide lyases (PLs), five auxiliary oxidoreductases (AAs) and 58 carbohydrate-binding modules (CBMs). Among these, GH2 (8.14 ± 0.41%) was the dominant gene, followed by GH3 (4.26 ± 0.09%), GH2 (4.10 ± 0.17%), GH43 (3.84 ± 0.32%), GH13 (3.71 ± 0.17%) and GT4 (3.02 ± 0.13%) genes. And in the GF vs. TMR comparison, 26 were enriched in the rumen of GF calves (9GH, 10GT, 2CE and 5CBM) and 26 were also enriched in the rumen of TMR group (16GH, 6CE, 3CBM and 1PL) (Figure 5A); in the GFF vs. TMR comparison, 19 were enriched in the GFF group (9GT, 8GH and 2CE), while 13 were enriched in the TMR group (6GH, 5CE, 1PL and 1CBM) (Figure 5B). Rumen microbial degraded amylum, mainly including EC 3.2.1.1 (α-amylase), EC 2.4.1.1 (starch phosphorylase), EC 3.2.1.3 (glucoamylase), EC 3.2.1.68 (isoamylase) and EC 3.2.1.41 (pullulanase) key enzymes, and the relative abundance of the EC 3.2.1.4 (cellulase), EC 3.2.1.91 (cellobiohydrolase) and EC 3.2.1.21 (β-glucosidase) for cellulose degradation decreased significantly in GF and GFF calves compared to TMR group (*p* < 0.05, Figure 5C).

### 3.5. The Relationship between the Types of Rumen Microorganisms and Their Functions

As the fermentation and digestion ability of feed nutrients is an important criterion for rumen development, the study further focused on the function of lipid metabolism and carbohydrate metabolism of rumen microorganisms. It was found that pyruvate metabolism was significantly enriched in GF (*p* < 0.05), C5 branched dibasic acid metabolism and propanoate metabolism were significantly enriched in GFF (*p* < 0.05), while pentose and gluconate interconversion and amino sugar and nucleoside sugar metabolism were significantly enriched in the rumen of TMR group (*p* < 0.05, Appendix A). In the secondary function of Lipid metabolism, Glycolipid metabolism, Glycophospholipid metabolism, Primary bile acid biosynthesis and Secondary bile acid biosynthesis were significantly enriched in GF and GFF (*p* < 0.05), and Arabic acid metabolism was significantly enriched in TMR (*p* < 0.05). Interestingly, the two pathways involved in fatty acid metabolism, which were “Fatty acid biosynthesis” and “Fatty acid degradation”, were enriched in the rumen of the TMR group, while Ether lipid metabolism, Synthesis and degradation of ketone bodies were enriched in the rumen of GF and GFF groups. Furthermore, compared with the GF group, Fatty acid biosynthesis was significantly enriched in the rumen of calves in the GFF group (*p* < 0.05, Appendix A). Notably, in the pyruvate metabolic pathway, the abundance of genes encoding enzymes also changed significantly. The 2.7.1.40 (phosphoenolpyruvate carboxylase) was significantly up-regulated in GF and GFF calves compared with the TMR group, while 2.7.9.1 (pyruvate-phosphate dikinase) was significantly down-regulated (*p* < 0.05, Figure 6).

By constructing the correlation analysis between bacterial species with lipids and carbohydrate metabolism. Forty-one (41) species were significantly correlated with amino sugar and nucleoside sugar metabolism and pentose and gluconate interconversion (R > 0.5 and *p* < 0.05, Appendix A). In the secondary functional pathway of lipid metabolism, 59 bacteria were significantly correlated with bile acid biosynthesis (R > 0.5 and *p* < 0.05), of which 38 were negatively correlated with bile acid biosynthesis. Moreover, a total of 43 species were significantly associated with two fatty acid pathways (R > 0.5 and *p* < 0.05), of which 22 bacteria were positively correlated with fatty acid biosynthesis (Appendix A). Among the 22 bacterial species positively associated with fatty acid biosynthesis, the five bacteria with the strongest correlation were *Prevotellaceae bacterium HUN156*, *Ruminococcus* sp. *NK3A76*, *Treponema berlinense*, *Ruminococcus flavefaciens* and *Succinimonas amylolytica*.

### 3.6. Analysis of Serum Metabolomics

The PLS-DA pattern recognition method was used to identify the overall metabolic differences between each pair of groups. The results showed significant differences in metabolites between each pair of groups. The model’s quality parameters were R2 > Q2 and Q2 < 0 (Figure 7A–C), indicating that the PLS-DA model was stable and reliable without overfitting. In addition, PCA analysis showed that the serum metabolites of the three groups of GF, GFF and TMR were well separated (Figure 7D), and the QC samples were tightly bound, indicating that the reproducibility of this experiment was good.

A total of 121 significantly differential metabolites were identified in the GF and TMR groups with VIP score > 1, Fold Change > 1.5 (*p* < 0.05), and 22 metabolites were classified as lipids and lipid-like molecules (Appendix A). Compared with the TMR group, the level of 9 lipids and lipid-like molecules was significantly decreased in the GFF group (Appendix A). Furthermore, regarding the key significant metabolites in the GF group, the level of 7 lipids and lipid-like molecules (including Androsterone, Palmitoylcarnitine and 4-Methylvaleric Acid, etc.) were significantly decreased in the GF group compared to those in the GFF group (Appendix A). Most of the differential metabolites in the GF and TMR groups were enriched in the vitamin digestion and absorption and bile secretion pathways by KEGG enrichment analysis (Figure 8A,B). Notably, As shown in Figure 8C, significant changes in Steroid hormone biosynthesis occurred in the GF group compared to the GFF calves (*p* < 0.05).

### 3.7. Correlation Analysis of the Metabolites and Microorganisms

Pearson correlation analysis was performed on the top 30 microbial taxa in relative abundance and 24 lipids and lipid-like modules metabolites (Figure 9). The results showed a significant correlation between 212 pairs (29.44%) out of 720 pairs (*p* < 0.05). Among these pairs, 95 pairs (44.81%) were negatively correlated, and 117 (55.19%) were positively correlated. Most of the microbes that were significantly associated with lipids and lipid-like molecule metabolites belong to the genus *Prevotella* and *Ruminococcaceae.*

## 4. Discussion

Through comprehensive analysis of phenotypic indexes, rumen metagenome and serum metabolism of Chinese Holstein bull calves, the study preliminarily explored the microbial and host metabolism molecular mechanism of interaction possibly affecting the animal growth and rumen development of bull calves. It evaluated the structural composition of rumen microorganisms under different feeding patterns and the effects of host metabolism on bull calves’ individual growth and rumen development, which selected the most suitable feeding pattern for calves from a molecular perspective.

The stability of rumen pH is the premise of rumen microbial function and the guarantee for the normal function of the rumen [38]. The rumen pH range of 6.0~7.0 was suitable for rumen nutrient digestion and absorption [39], when pH < 6.5, is not conducive to the digestion of cellulose, and the buffering capacity of the rumen is significantly reduced. When pH < 6.0, the activity of cellulose decomposition bacteria is inhibited or even killed, and the buffering capacity of the rumen is significantly reduced [40]. The study’s results on rumen pH value indicated that the digestibility of GF group calves was the strongest, followed by TMR group calves, and GFF group calves were the weakest. Although the rumen fluid pH has no direct impact on the development of the rumen, Krehbiel et al. [41] found that it could change the proportion of butyric acid in the rumen fluid to cause changes in the proportion of SCFA, which was the best stimulus for the development of the rumen. The absorption of butyric acid also increased with the decrease in pH [42]. Simultaneously, VFA in the rumen was an essential component in maintaining the growth and reproduction of ruminants, which could meet about 70% of the energy demand [43,44,45], so the concentration of VFA was an important reference index for rumen fermentation. The results of the study showed that the concentrations of short-chain fatty acids such as acetic acid, propionic acid, and butyric acid in the rumen of three groups of calves were significantly different, indicating that the abundance of VFA-related microorganisms in the rumen of calves was changed under different feeding patterns, further affecting the content of VFA in the rumen. Particularly, the A/P (acetic acid to propionic acid ratios) in the rumen of the GF group calves was significantly reduced compared with the TMR group, indicating that the calves with GF feeding pattern were more conducive to rumen fermentation, increasing the yield of propionic acid and reducing the ethylene propylene ratio, which was beneficial to improving feed utilization, similar to the research results of Kang et al. [46].

Bacteria are the most abundant microbial community in the rumen of dairy cows, which is consistent with the results found in previous studies [47]. In the study, the difference in rumen microorganisms among GF, GFF and TMR calves was mainly shown in bacteria. The dominant phylum in the rumen of calves were Bacteroidetes, Firmicutes and Proteobacteria, and the dominant genus was *Prevotella*, which is consistent with the results of Jami et al. [48] analysis of dominant microorganisms in the rumen of Holstein cows. It was found that bacteria were the main participants in the degradation and fermentation of most macromolecular substances in feed, indicating that bacteria played an important role in promoting the growth and development of the host rumen than archaea, viruses, and eukaryotes. The results showed that the relative abundance of *Prevotella multisaccharivorax* and *Prevotella* sp. *AGR 2160* were significantly increased in the rumen of GF and GFF calves compared to the TMR group. *Prevotella* might improve the glucose metabolism of the host by promoting increased glycogen storage [49], *Prevotella multisaccharivorax* fermented sugars (including glucose, glycerol, lactose, maltose, etc.). It also variably fermented arabinose and salicin, and the products of fermentation are acids; two of the significant products formed are acetic acid and succinic acid [50], which indicated that the rumen fermentation ability of GF and GFF groups of calves for carbohydrates in feed was stronger. Notably, at the species level, most of the species that showed higher abundance in the rumens of the three groups of calves belonged to the *Prevotella* genus. This genus could produce succinic acid and acetate from starch and proteins in feed and was one of the core bacteria in the rumen of dairy cows [51].

Regarding archaea, *Metanobrevibacter*, *Metanosphaera* and *Metanolacia* had high relative abundance in GF and GFF groups, which could use hydrogen, carbon dioxide and formate as the substrate for methane production [52], and use hydrogen in the rumen anaerobic environment as the final way of carbohydrate fermentation, so that microorganisms involved in fermentation could play the best function and support the complete oxidation of the substrate. If hydrogen for carbohydrate fermentation is not removed, it might inhibit the metabolism of rumen microorganisms [53]. This meant that rumen fermentation of calves with GF and GFF feeding patterns produced more hydrogen, which increased the consumption of carbohydrates and provided more energy [54].

Previous research has shown that GTs were responsible for catalyzing the glycosidic bond cleavage and were the second abundant CAZy family (accounting for 19–24% of the total CAZymes) [55]. CBMs didn’t have enzymatic activity, but they could help Cazymes (such as GHs, CEs, and AAs) combine with polysaccharides to enhance their activity [56]. In the study, GF and GFF calves had higher GT abundance than TMR calves, indicating that the rumen of calves with these two feeding patterns was more conducive to the phosphorylation of carbohydrate nutrients, which was helpful to the digestion of carbohydrates. In combination with the abundance changes of key enzymes that degrade starch and cellulose, the study speculated that the GF group increased the amount of the binding activity reduces the content of starch and cellulose in the rumen, which reduced the gene abundance of glucoamylase (EC 3.2.1.3), cellulase (EC 3.2.1.4), cellobiohydrolase (EC 3.2.1.91) and β-The gene abundance of glucosidase (EC 3.2.1.21) in the rumen.

An important function identified in the study was pyruvate metabolism. For ruminants, pyruvate was an intermediate product of carbohydrate fermentation by rumen microorganisms [57], and the ATP released by the conversion of pyruvate into volatile fatty acids (VFAs) in the rumen could promote the growth of rumen microorganisms. The results showed that pyruvate metabolism and glycolysis/gluconeogenesis pathway were enriched in the rumen of GF calves. Phosphoenolpyruvate (PEP) and pyruvate could be transformed mutually, and this step was not only an important part of the pyruvate metabolic pathway but also the last step of the glycolysis pathway [58]. In previous studies, dietary supplementation of calcium pyruvate increased rumen VFAs content, reduced dietary protein degradation, and reduced rumen NH_3_-H, suggesting that calcium pyruvate was beneficial to rumen metabolism [59,60]. In pyruvate metabolism, the down-regulation of pyruvate-phosphate dikinase (EC: 2.7.9.1) and the up-regulation of phosphoenol-pyruvate carboxylase (EC: 4.1.1.31) promoted the last step of the glycolytic pathway (conversion of PEP to pyruvate), and a large amount of ATP was produced, which could promote rumen microorganisms Valine, leucine, and isoleucine biosynthesis. The results indicated that calves could promote rumen microbial protein synthesis under GF feeding mode [61], improve the utilization rate of rumen nitrogen and promote the rumen fermentation efficiency of calves. Additionally, the results showed that the species with a significant positive correlation with the pyruvate metabolic pathway belong to the *Clostridiales* and the *Methanobrevibacter* genus, which indicated that *Clostridiales* and *Methanobrevibacter* play an important role in pyruvate metabolism.

Besides carbohydrate metabolism, the functions of rumen microorganisms related to lipid metabolism also have important physiological roles in rumen development [62,63,64,65]. Therefore, the study further focused on the lipid metabolism function of rumen microorganisms. It was found that Arachidonic acid metabolism was significantly enriched in the rumen microbiota of TMR group calves. Arachidonic acid metabolism was an important part of the inflammatory cascade reaction, which could lead to the production of prostaglandins and leukotrienes, most of which were strong proinflammatory factors [66,67,68]. And it might cause abnormal secretion of gastric acid and gastrointestinal hormones, leading to insufficient gastrointestinal motility, and extremely prone to feeding intolerance [69], which indicates that feeding the TMR diet to calves might cause and aggravate rumen inflammation reaction, thereby inhibiting the development of rumen, reducing the fermentation capacity of rumen, and leading to significant reduction in weight gain of calves in TMR group. Bile acid was a major component of bile, which could be secreted into the gastrointestinal tract after eating to promote the digestion and absorption of triacylglycerol, cholesterol and fat-soluble vitamins [70], and it was essential to maintain a healthy gastrointestinal microbiota, balance lipid and carbohydrate metabolism, insulin sensitivity and innate immunity [71]. BAs had an emulsifying effect on dietary fat, increasing the surface area of fat to make it easier to be digested by trypsin [72]. Some essential vitamins (A, D, E and K) were non-polar lipids and could be absorbed by the body only with the participation of BAs [73,74].

Additionally, the bile acid cycle plays a key role in physiological cholesterol homeostasis [75]. In the study, the primary and secondary bill acid biosynthesis were significantly enriched in GF and GFF groups, indicating that GF and GFF calves might have a stronger emulsifying effect on the fat in the feed, which was more conducive to the digestion and absorption of lipids, and provide nutrients needed for the host organism metabolism. Interestingly, the two pathways related to fatty acid metabolism were enriched in the rumen of TMR calves. Previous studies have found that fatty acids were precursors of the synthesis of inflammatory mediators, which played an important role in regulating innate immunity and might contribute to the development of some autoimmune diseases [76,77,78]. Indicated that feeding TMR to calves might cause rumen inflammation, which was also consistent with the previous analysis results.

Untargeted metabolomics analysis of serum showed significant differences in metabolites among GF, GFF and TMR groups. Compared with the TMR group, Hexanoylcarnitine was significantly decreased in GF and GFF calves, and Propionylcarnitine was significantly increased in GF calves. Carnitine is a type B fatty acid-produced process of fatty acid esterification, which plays an important role in regulating muscle oxidative metabolism [79]. Caprylcarnitine and propionyl carnitine are the long-chain acylcarnitines (LCACs) fatty acid derivatives of carnitine [80]. The accumulation of most LCACs during fatty acid oxidation might lead to insulin resistance [81]. In the study, the level of Hexanoylcarnitine in TMR calves was significantly higher than that in GF and GFF calves, which might lead to the disturbance of fatty acids and glucose circulation in the body, and further inhibit the growth rate of calves. Concomitantly, the level of Propionylcarnitine in the organism might be a sign of cobalamin deficiency and had beneficial effects on heart metabolism and blood circulation [82]. Propionylcarnitine had a higher level in GF calves, indicating that the metabolism of GF calves was more vigorous. KEGG enrichment of differential metabolites showed that most of the differential metabolites of GF and GFF were enriched in Bile secret pathways, and the related metabolite Ursodeoxic acid was significantly up-regulated in the GF calf group. Ursodeoxycholic acid was formed by the transformation of deoxycholic acid, which helped regulate lipid metabolism and the integrity of the intestinal barrier [83]. And bile acids could be used as a signal molecule to participate in energy metabolism and other regulatory processes by activating bile acid receptors [84]. Thus, the growthiness of calves might be related to the activation of bile acid receptors in the organism under the GF feeding pattern.

Importantly, it was found that there was a very close relationship between rumen microorganisms and lipid metabolites. Most of the microorganisms that have a significant correlation with most lipid metabolites belong to the *Prevotella* and *Ruminococcaceae* genus. *Prevotella* was beneficial to microbial metabolism to produce short-chain fatty acids [85,86], which could protect the gastrointestinal mucosal barrier, regulate immune function [87,88], and alleviate the effects of rumen structure and metabolic disorders and immune disorders caused by the feeding process. *Ruminococcaceae* was the main cellulolytic bacteria in the rumen, which was related to the release of inflammatory and cytotoxic factors and could participate in immune regulation and maintain a healthy homeostasis environment [89]. It was a potentially beneficial bacterium. Some studies showed that microorganisms could act as regulators of lipid digestion, absorption, storage and secretion in the gastrointestinal tract [90,91,92], which indicated that their synergistic action with the host could hydrolyze lipid substances more efficiently, which was conducive to the absorption of fat by the organism, and was of great significance to the growth and development of calves.

The study further demonstrated that feeding only concentrated feed to calves could cause a series of inflammatory reactions, lead to immune system disorders, and affect the normal growth and development of the rumen, but there were certain differences from previous research results at the genetic level [93]. Although the study found reliable results in rumen microorganisms body metabolism, due to the small number of test samples and the lack of relevant data on rumen microbial metabolism, further research will be needed in the individual growth, rumen development and disease of calves.

## 5. Conclusions

Calves achieved similar feeding results with “concentrate plus hay”, “concentrate” and “TMR” feeding patterns, but the rumen pH of calves was more suitable for the function of microorganisms with “concentrate plus hay” feeding patterns. The composition of microorganisms in “concentrate plus hay” feeding pattern calves was more helpful to the digestion of carbohydrates and alleviated inflammation. Simultaneously, the synergistic effect with the host could hydrolyze lipid substances more efficiently and promote the absorption of fat by the organism, which was of great significance to the growth of calves. Due to the small sample size of this study, further research should increase the sample size of each group and focus on the interaction between rumen microorganisms, microbial metabolism, and host metabolism.

## Figures and Tables

**Figure 1 microorganisms-11-02423-f001:**
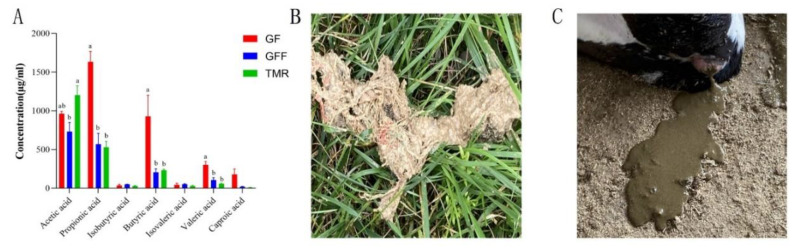
Overview of rumen contents in calves. (**A**) The concentration of sCFA in rumen fluid and different lowercase letters indicated significant differences between groups (*p* < 0.05). (**B**) “Hair bulbs” in the rumen of GFF calves. (**C**) Calves in the TMR group had slight diarrhea.

**Figure 2 microorganisms-11-02423-f002:**
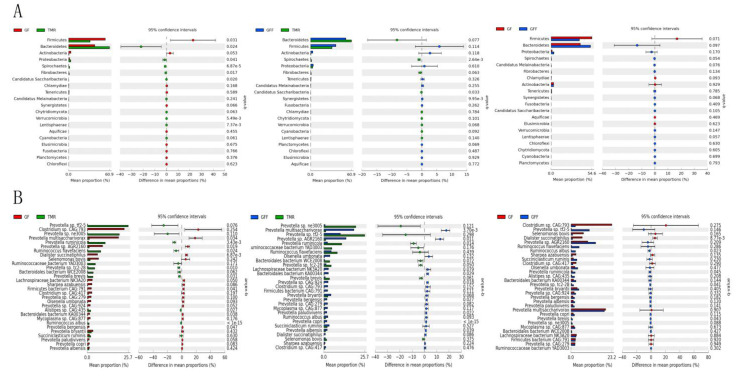
Comparison of microbial in rumen bacteria among the three groups. (**A**) Comparison of bacteria at the phylum level. (**B**) Comparison of bacteria at the species level.

**Figure 3 microorganisms-11-02423-f003:**
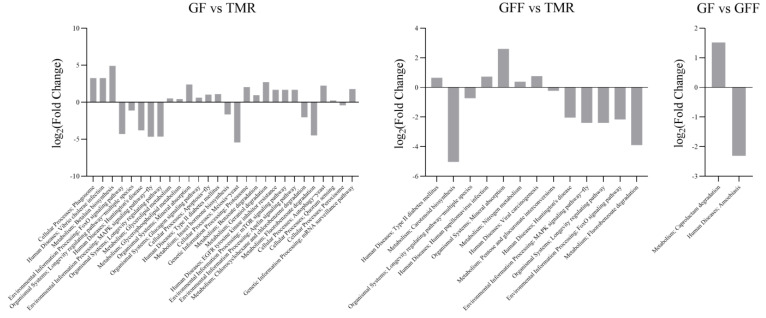
The log_2_FC of extremely significantly enriched metabolic pathways among the three groups, *p* < 0.01.

**Figure 4 microorganisms-11-02423-f004:**
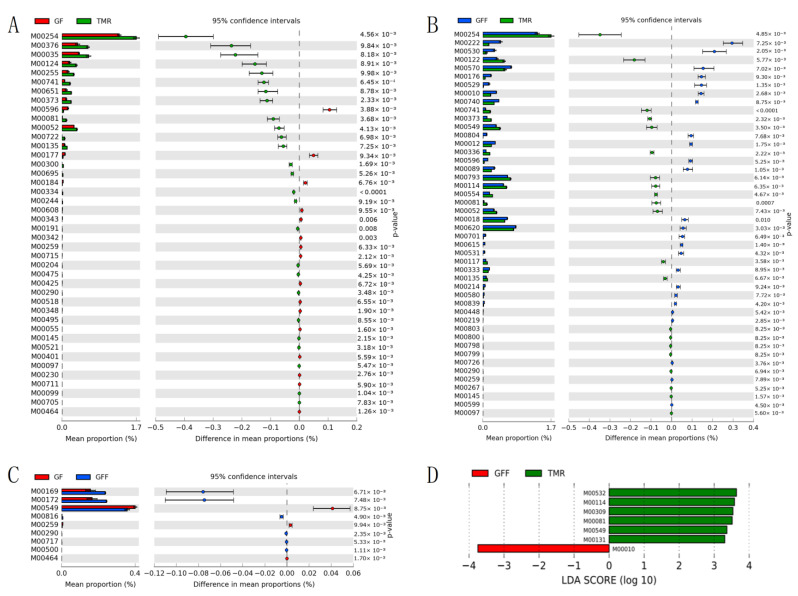
Comparison of rumen microbial extremely significantly different KEGG modules among the three groups. (**A**) Significantly different modules between GF and TMR calves. (**B**) Significantly different modules between GFF and TMR calves. (**C**) Significantly different modules between GF and GFF calves. (**D**) Carbohydrate metabolism. Significant different pathways were tested by Linear discriminant analysis effect size (LEfSe) analysis with linear discriminant analysis (LDA) score of >3 and *p* < 0.05.

**Figure 5 microorganisms-11-02423-f005:**
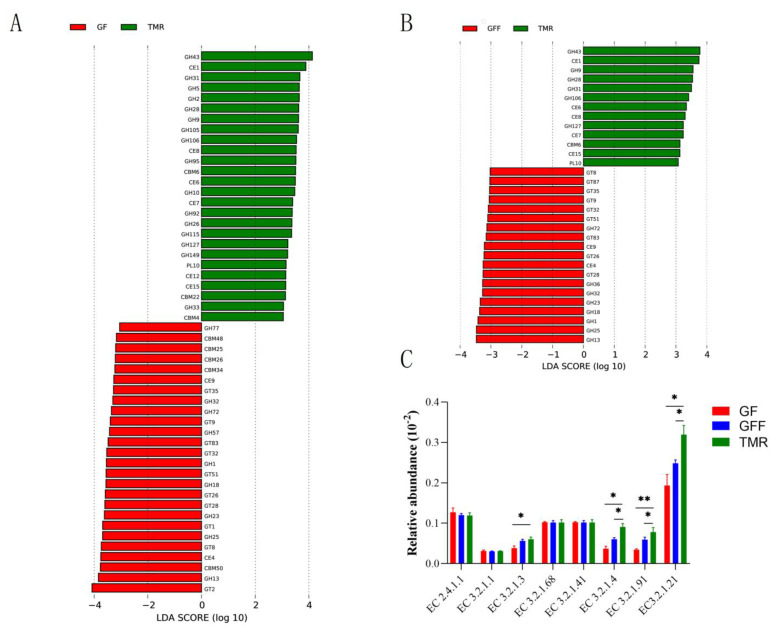
Differential CAZyme functions among three groups. (**A**,**B**) Significantly different CAZyme functions in GF and GFF groups compared with the TMR group, significantly different CAZymes were tested by Linear discriminant analysis effect size (LEfSe) analysis with linear discriminant analysis (LDA) score of >3 and *p* value of <0.05. (**C**) Comparisons of the gene abundance of KO enzymes for amylase and cellulase, * significantly different among groups, * *p* < 0.05, ** *p* < 0.01.

**Figure 6 microorganisms-11-02423-f006:**
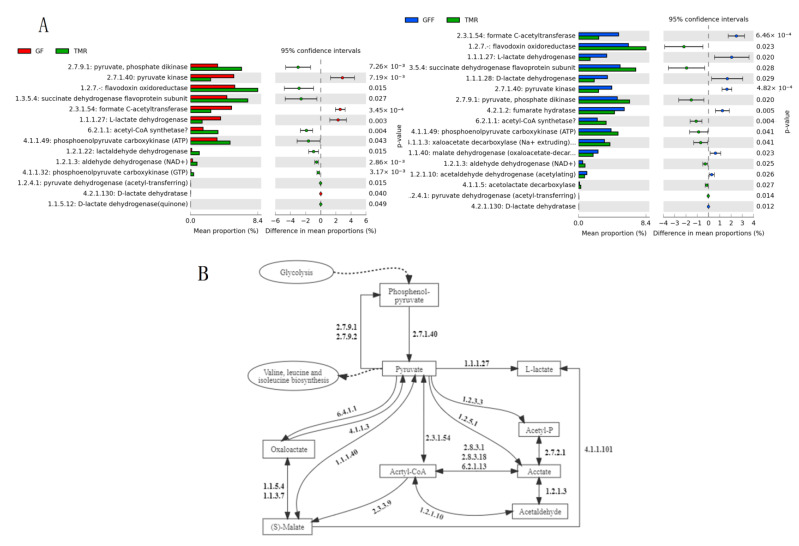
Differential ECs and ECs involved in pyruvate metabolism (EC indicated the number of Enzyme in the pathway). (**A**) The ECs of calves in the GF and GFF groups were significantly different from the TMR group. (**B**) Microbial functions and species involved in Pyruvate Metabolism in the rumen of calves.

**Figure 7 microorganisms-11-02423-f007:**
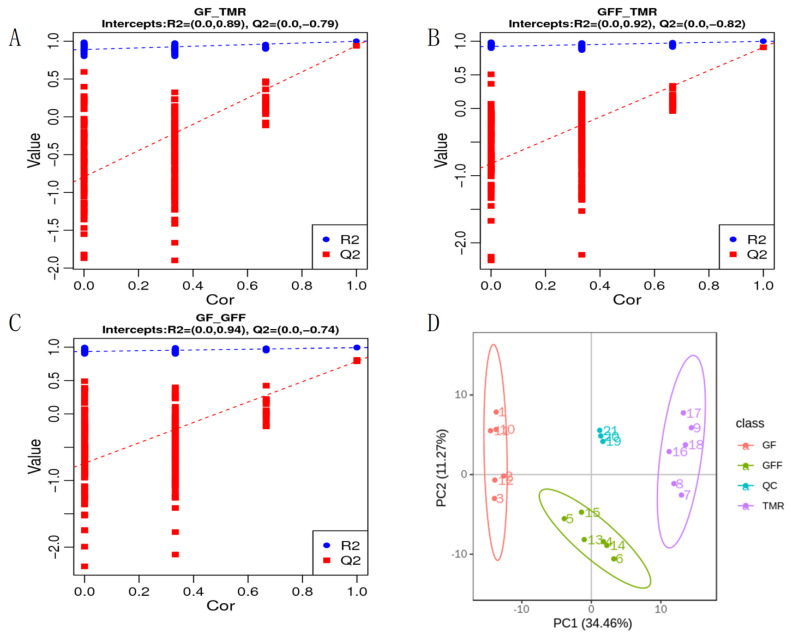
Multivariate statistical analysis of metabolomic. (**A**) PLS−DA score plots for GF and TMR groups. (**B**) PLS−DA score plots for GFF and TMR groups. (**C**) PLS−DA score plots for GF and GFF groups. (**D**) PCA score plots for serum samples in the three groups. Each point represents an individual sample.

**Figure 8 microorganisms-11-02423-f008:**
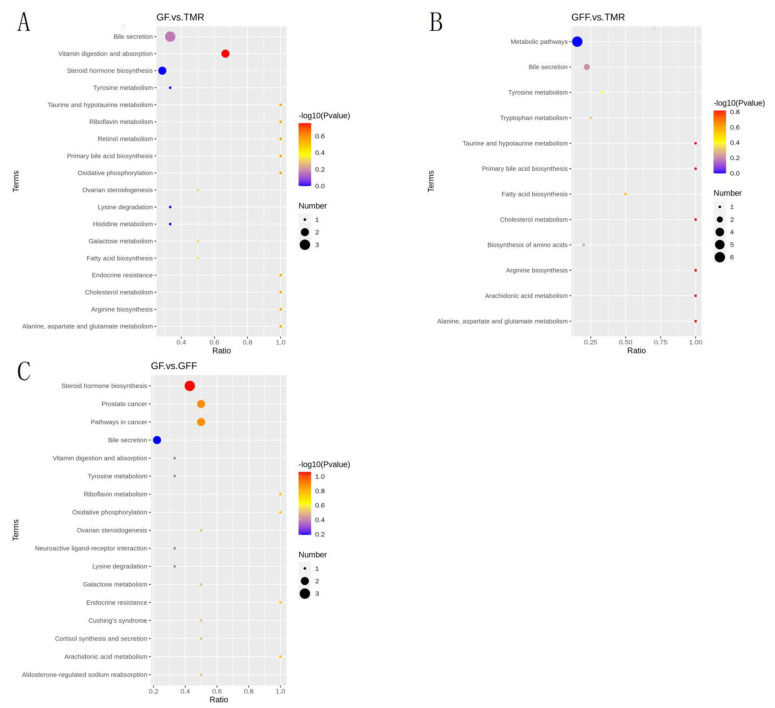
KEGG enrichment analysis of differential metabolites between the three groups. (**A**) KEGG enrichment analysis of differential metabolites between GF and TMR groups. (**B**) KEGG enrichment analysis of differential metabolites between GFF and TMR groups. (**C**) KEGG enrichment analysis of differential metabolites between GF and GFF groups.

**Figure 9 microorganisms-11-02423-f009:**
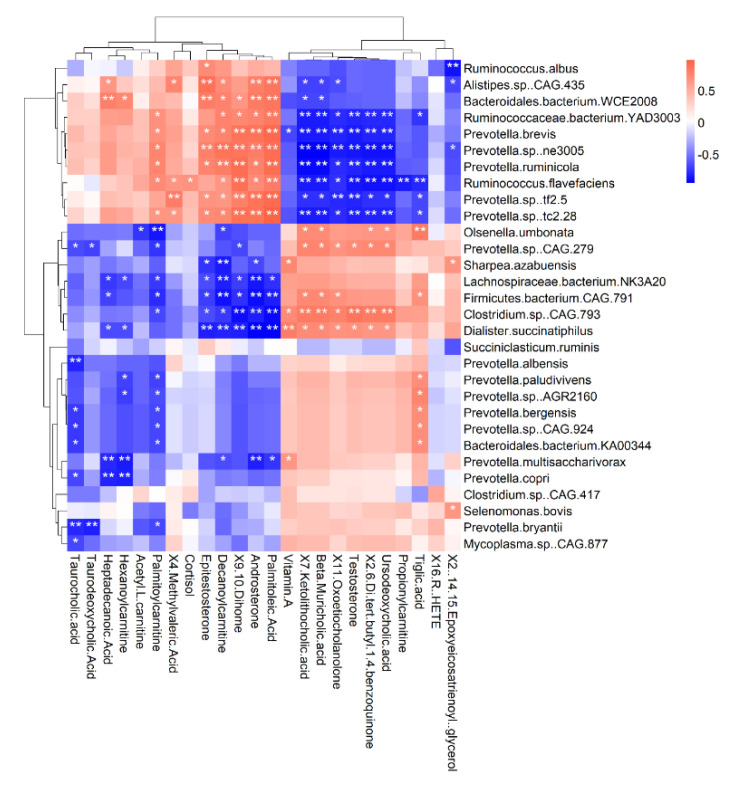
Correlation analysis between the significantly differential metabolites and rumen bacterial taxa identified * *p* < 0.05, significant correlation, ** *p* < 0.01, extremely significant correlation.

**Table 1 microorganisms-11-02423-t001:** Growth performance and rumen development parameters of calves in GF, GFF and TMR Groups.

Parameters	GF	GFF	TMR
birth weight (kg)	41.38 ± 1.37	40.95 ± 1.58	42.45 ± 1.63
slaughtering weight (kg)	100.62 ± 1.94 ^a^	100.07 ± 1.92 ^a^	90.77 ± 2.42 ^b^
liveweight gain (kg)	59.23 ± 7.77	59.12 ± 8.44	48.32 ± 11.82
weight of rumen (kg)	3.35 ± 0.11 ^a^	2.35 ± 0.69 ^b^	2.30 ± 0.0.68 ^b^
volume of rumen (L)	32.62 ± 0.41 ^a^	20.48 ± 3.22 ^b^	21.00 ± 6.07 ^b^
The ratio of rumen to live weight (%)	2.14 ± 0.03 ^ab^	2.35 ± 0.43 ^a^	1.57 ± 0.17 ^b^
The ratio of rumen to total stomach weight (%)	60.95 ± 2.40 ^a^	61.13 ± 3.34 ^a^	52.38 ± 0.59 ^b^
the pH of the rumen	6.63 ± 0.16 ^a^	5.33 ± 0.26 ^b^	6.30 ± 0.13 ^a^

Note: There was no same lowercase letter in the same row of data shoulder mark to indicate that the difference was significant (*p* < 0.05), and the same letter or no letter indicated that the difference was not significant (*p* > 0.05), GF: “Concentrate” feeding pattern, GFF: “Concentrate + hay” feeding pattern, TMR: “Total Mixed Rations” feeding pattern.

**Table 2 microorganisms-11-02423-t002:** Composition of rumen microorganisms of calves.

Sample	Archaea	Bacteria	Eukaryota	Viruses	Others
GF1	545,056	35,433,043	3710	25,237	6,172,506
	1.292%	84.005%	0.009%	0.060%	14.634%
GF2	460,365	35,799,595	2891	43,392	5,873,309
	1.091%	84.874%	0.007%	0.103%	13.925%
GF3	363,255	34,506,676	1810	172,246	7,135,566
	0.861%	81.809%	0.004%	0.408%	16.917%
GFF1	221,641	35,363,072	2490	60,795	6,531,555
	0.525%	83.839%	0.006%	0.144%	15.485%
GFF2	998,998	34,991,380	2715	43,146	6,143,314
	2.368%	82.958%	0.006%	0.102%	14.565%
GFF3	368,137	35,470,400	6350	55,213	6,279,452
	0.873%	84.094%	0.015%	0.131%	14.887%
TMR1	76,604	36,590,887	5652	31,771	5,474,639
	0.182%	86.750%	0.013%	0.075%	12.979%
TMR2	158,477	36,270,649	11121	40,252	5,699,055
	0.376%	85.991%	0.026%	0.095%	13.511%
TMR3	215,850	35,870,047	14,373	80,548	5,998,736
	0.512%	85.041%	0.034%	0.191%	14.222%
mean	378,709	35,588,416	5679	61,400	6,145,348
	0.898%	84.374%	0.013%	0.146%	14.569%
SEM	92,075.71	211,140	1449.27	14,898.98	162,144.56
	0.218%	0.501%	0.003%	0.035%	0.384%

**Table 3 microorganisms-11-02423-t003:** Relative abundances of the top 10 components of the microbiota at the phylum level.

Phylum	Relative Abundance (%)			Mean	SEM
	GF	GFF	TMR		
Firmicutes	43.68	29.69	25.61	32.99	0.0323
Bacteroidetes	31.11	41.38	48.89	40.46	0.0302
Proteobacteria	10.70	3.08	2.44	21.96	0.0047
Actinobacteria	2.72	2.56	0.44	19.10	0.0053
Chlamydiae	0.32	0.17	0.19	0.23	0.0003
Tenericutes	0.56	0.62	0.44	0.54	0.0008
Euryarchaeota	1.08	1.25	0.35	0.89	0.0022
Spirochaetes	0.11	0.37	1.13	0.54	0.0016
Fibrobacteres	0.08	0.23	0.68	0.33	0.0010
Synergistetes	0.07	0.14	0.03	0.08	0.0002

## Data Availability

All the figures and tables used to support the results of this study are included.

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
