# Peer review of "A Preliminary Study of the Potential Molecular Mechanisms of Individual Growth and Rumen Development in Calves with Different Feeding Patterns"

_microorganisms, 2023, doi:10.3390/microorganisms11102423_

Round 1

Reviewer 1 Report (Previous Reviewer 4)

The version that the author submitted still contains automatic revisions to be deleted

Line 178-182

Line 211

Line 260

Line 575

Need to improve the resolution of the figures (the figures are invisible)

The version that the author submitted still contains automatic revisions to be deleted

Line 178-182

Line 211

Line 260

Line 575

Need to improve the resolution of the figures (the figures are invisible)

Author Response

Thank you very much for your suggestion. We have removed the 'deleted automatic revision' from the manuscript,and also increased the resolution of the images (mainly in Fig.2, Fig.5, and Fig.88). Thank you again for taking the time out of your busy schedule to review my manuscript. Good luck.

Reviewer 2 Report (Previous Reviewer 3)

The authors made the suggested changes and the manuscript improved and is now ready to be published.

Author Response

Thank you very much for taking the time out of your busy schedule to review my manuscript. Your feedback has been very helpful in improving the quality of my manuscript. Thank you again and best wishes

Reviewer 3 Report (Previous Reviewer 2)

Authors (Kaisen Zhao, Jie Wang, Mianying Li, Huimei Fan, Meigui Wang, Siqi Xia, Yang Chen, Xue Bai, Zheliang Liu, Jiale Ni, Wenqiang Sun, Xianbo Jia, and Songjia Lai) have taken into account all of my comments provided in my review dated July 4, 2023, and have improved the manuscript accordingly for submission to the journal Microorganisms (ISSN 2076-2607) in its present form.

Sincerely, reviewer.

Author Response

Thank you very much for taking the time out of your busy schedule to review my manuscript. Your feedback has been very helpful in improving the quality of my manuscript. Thank you again and best wishes

This manuscript is a resubmission of an earlier submission. The following is a list of the peer review reports and author responses from that submission.

Round 1

Reviewer 1 Report

I don't see the reasons to change my recommendation since the author's future research plan in their response doesn't address or resolve the issues in my comments 1 and 3.

Moderate editing of English language required.

Reviewer 2 Report

The authors took into account all of my comments submitted on July 4th and made corrections to the publication.

Reviewer 3 Report

The manuscript by Wang et al. shows the results of a very interesting study on the effect of diet on the ruminal microbiota. But it needs major corrections.

1. Requires a rewrite with careful revision of the English.

2. Always the initials must be placed the first time they are used.  Although the terms were of frequent use.

3. At the end of the introduction, the objective is mentioned to study the effect of diets on rumen development and calf growth, in materials and methods it is not explained what methodology will be used to achieve this objective, but then they appear in results. These results speak of rumen weight and volume, parameters that determine growth but not rumen development. On the other hand, the use of non-relative values ​​in relation to body weight downplays these parameters. In general, the experimental design is unclear, it would be interesting to present a summary of it.

4. The amount of tables and graphs is excessive. Those that do not present results that are later discussed should be reduced and placed as supplementary material.

5. Accompanying this reduction, the size of those graphs and tables that remain in the text must be increased.

6- In the discussion, some aspects are complex and seem unfounded, for example those related to inflammation and bile production, which are based too much on arguing with results from species with a very different digestive system and metabolism.

7. The discussion is very long and difficult to understand; I think it should be rewritten highlighting the main aspects found.

The manuscript requires a rewrite with careful revision of the English.

Reviewer 4 Report

General comment

This is an interesting work to publish in the microorganisms but after a minor revision.

the idea of studying the relationship between the feeding system and rumen microorganisms of calves is interesting and deserves to be published in the MDPI database.

the results of this work will undoubtedly have a positive impact on animal production to understand and control the growth performance of calves.

must check the version submitted for publication I believe that the author submitted the draft version with erasures.

Specific comment

Abstract

Line 24 : Please define the abbreviation before it use.

Line 26-29 : Delete this part

Introduction

It's necessary to review the abbreviations throughout the manuscript

Materials and Methods

Line 107 : ad libitum instead  ad libitum.

Line 123 : determined instead measured

Results

Line 203-204 : acetic, propionic and butyric acids instead acetic acid, propionic acid and butyric acid

Table 1: parameters instead Items

Table 1 define the abbreviation

Line 330-335 : please improve the quality of Figure 2 and 3

The English should be checked by an English speaker